# Ubiquitin and Ubiquitin-like Proteins in Cancer, Neurodegenerative Disorders, and Heart Diseases

**DOI:** 10.3390/ijms23095053

**Published:** 2022-05-02

**Authors:** Jin-Taek Hwang, Ahyoung Lee, Changwon Kho

**Affiliations:** 1Food Functionality Research Division, Korea Food Research Institute, Jeonju 55365, Korea; jthwang@kfri.re.kr; 2Department of Food Biotechnology, University of Science & Technology, Daejeon 34113, Korea; 3Institute of Korean Medicine, School of Korean Medicine, Pusan National University, Yangsan 50612, Korea; ahyoung.lee@pusan.ac.kr; 4Division of Applied Medicine, School of Korean Medicine, Pusan National University, Yangsan 50612, Korea

**Keywords:** post-translational modification, ubiquitin, ubiquitin-like proteins, ubiquitin-proteasome system, disease association and progression

## Abstract

Post-translational modification (PTM) is an essential mechanism for enhancing the functional diversity of proteins and adjusting their signaling networks. The reversible conjugation of ubiquitin (Ub) and ubiquitin-like proteins (Ubls) to cellular proteins is among the most prevalent PTM, which modulates various cellular and physiological processes by altering the activity, stability, localization, trafficking, or interaction networks of its target molecules. The Ub/Ubl modification is tightly regulated as a multi-step enzymatic process by enzymes specific to this family. There is growing evidence that the dysregulation of Ub/Ubl modifications is associated with various diseases, providing new targets for drug development. In this review, we summarize the recent progress in understanding the roles and therapeutic targets of the Ub and Ubl systems in the onset and progression of human diseases, including cancer, neurodegenerative disorders, and heart diseases.

## 1. Introduction

Post-translational modification (PTM) is a key mechanism that precisely regulates protein activity to determine cellular metabolism or fitness in response to genetic and environmental changes. PTMs mediated by ubiquitin (Ub) and various ubiquitin-like proteins (Ubls) can alter the functions, locations, and levels of their specific targets [1]. These modifiers are covalently bound to their targets via an enzymatic cascade involving three enzymes, E1 (activating enzyme), E2 (conjugating enzyme), and E3 (ligase). E1 enzymes activate Ub/Ubl by catalyzing C-terminal adenylation of Ub/Ubl with the aid of adenosine triphosphate (ATP) to form a covalent thioester bond. E2 enzymes then accept the activated Ub/Ubl via a transthioesterification reaction. Finally, the Ub/Ubl-loaded E2 enzyme, either alone or with the help of a partner E3 enzyme, transfers Ub/Ubl to a specific target lysine residue, creating an amide (or isopeptide) bond between Ub/Ubl and the substrate. Deubiquitinating enzymes (DUBs) and Ub/Ubl-specific proteases (ULPs) are responsible for Ub deconjugation, editing, and recycling (Figure 1A).

Since Ub was first discovered in 1975, bioinformatic, genetic, and biochemical studies have revealed diverse sets of Ubls. Ub family proteins are structurally similar to ubiquitin (i.e., β-grasp globular fold structure) [2] and are divided into two classes: ubiquitin-like proteins (type I) and ubiquitin-like domain proteins (type II) (Figure 1B). Ubls (type I) have a characteristic sequence motif consisting of one or two glycine residues at the C-terminus, through which covalent attachment occurs. Almost all Ubls are conjugated to other proteins, except for at least one protein (e.g., ATG8, which attaches to lipids). The ubiquitin-like domain protein (type II) lacks a C-terminal diglycine motif and cannot be conjugated to a target. Instead, it can function as a protein-protein interaction domain. Ten Ubl orthologs that function as protein modifiers in humans have been identified [3]: ATG8, ATG12, FAT10, FUBI, HUB1, ISG15, NEDD8, SUMO, UFM1, and URM1. Although Ubls are structurally related to Ub, their biochemical roles are distinct because they are associated with DNA repair, nuclear transport, proteolysis, translation, autophagy, and immune responses.

In this review, we outline the role of Ub/Ubl modifications in disease-associated mechanisms, focusing on ubiquitination and SUMOylation. We also discuss current small-molecule drugs targeting these PTMs as therapeutic strategies for complex diseases, such as cancer, neurodegenerative disorders, and heart disease.

## 2. Ub/Ubl Modifications in Disease Pathogenesis and Treatment

### 2.1. Ubiquitin-Proteasome System

Ubiquitin-proteasome system (UPS)-mediated proteolysis involves two significant steps: (i) attachment of Ub to target proteins via ubiquitination, and (ii) degradation of Ub-proteins by the proteasome (Figure 2). The Ub molecule has seven acceptor lysines, each of which can participate in subsequent ubiquitination to generate multiple mono-Ub proteins or a poly-Ub chain. Different Ub linkages result in different conformations of the Ub chain and produce diverse molecular signals in cells [4]. Proteins homogenously linked to poly-Ub are mainly delivered to and degraded by the 26S proteasome (20S proteasome+19S regulatory particles), where Ub is recycled. The UPS is primarily responsible for the full or partial processing of most intracellular proteins (more than 80%) or stress response transcription factors (e.g., nuclear factor kappa-light-chain-enhancer of activated B cells (NF-κB) or hypoxia-inducible factor). Maladaptive UPSs with problems including impairment of the ubiquitination process, insufficient substrate delivery to the proteasome, or loss of proteasome activity may contribute to the progressive disruption of cellular proteostasis. In addition to proteolysis, ubiquitination is involved in non-proteolytic events, such as autophagy, inflammation, DNA repair, multi-protein complex assembly, and regulation of enzymatic activity. Abnormalities in this system have been implicated in the pathogenesis of numerous human diseases, and pharmacological manipulation of the UPS appears to alter the outcome of many diseases [5]. Therefore, the UPS is an important therapeutic target.

#### 2.1.1. Abnormal Ub Pathways in Cancer

In cancer cells, an abnormal UPS contributes to cell cycle progression, faulty DNA repair, apoptosis, angiogenesis, receptor downregulation, and gene transcription through the degradation of oncogenes or tumor suppressors. Notable examples include the downregulation of cell cycle-related and tumor suppressor proteins, such as p53 and cyclin-dependent kinase inhibitor 1 B (p27), and the upregulation of oncogenic proteins, such as the transcription factor NF-κB.

To date, only two Ub E1 enzymes have been identified: ubiquitin-like modifier-activating enzyme 1 (UBA1) and ubiquitin-like modifier-activating enzyme 6 (UBA6). These enzymes are essential for supporting cellular stress, including DNA damage and protein toxicity, in cancer cells. Bioinformatics approaches have identified E1 enzymes associated with the pathogenesis of human cancer [6]. The aberrant expression of UBA1, the predominant isoform of the two E1 enzymes, is associated with the pathogenesis of lung cancer [7] and cutaneous squamous cell carcinoma [8]. E1 enzyme activity is also known to increase in malignant hematologic cells [9].

The human genome encodes 38 Ub E2 enzymes. Elevated levels of E2-conjugating enzymes, which are associated with lower patient survival rates, have frequently been observed in various cancer specimens. For example, the upregulation of the ubiquitin-conjugating enzyme E2 L3 (UBE2L3) has been detected in non-small cell lung cancer (NSCLC) tissues compared to non-cancerous tissues. High expression of UBE2L3 is associated with advanced tumor stages and adverse outcomes [10]. Mechanistically, UBE2L3 catalyzes the ubiquitination and proteasomal degradation of p27 by interacting with the S-phase kinase-associated protein 2 (Skp2), a p27-specific E3 ligase.

The human UPS comprises more than 700 E3 ligases and DUBs, and the abnormal expression of these enzymes and the presence of functional mutations are associated with cancer progression. Mouse double-minute 2 homolog (MDM2), a major negative regulator of p53, and the p27 destabilizer Skp2 are well-studied E3 ligases because of their potential to cause human cancer. Ubiquitin-specific protease 7 (USP7) is the most widely studied DUB that modulates p53 function through Ub clearance. USP7 directly regulates p53 stability or downregulates p53 by stabilizing MDM2. USP7 is highly expressed in various malignancies [11].

Examples of UPS defects associated with cancer are summarized in Table 1.

#### 2.1.2. UPS Inhibitors Used to Treat Cancer

The clinical success of cancer treatments targeting proteasome inhibitors has demonstrated that UPS is a therapeutic strategy. In recent years, extensive studies of E1 activating enzymes as targets of pharmacological inhibition have identified potential novel targets. Targeting common pathways, such as E1 enzymes or the proteasome, may lead to non-specific effects, but can be applied to therapeutic and safety windows that will be useful for short-term therapy. Inhibiting single E3 Ub ligases or specific proteasome-associated DUB enzymes is another strategy for modulating the UPS, which is expected to increase specificity and lower toxicity.

Proteasome inhibitors: Proteasome inhibitor (PI)-induced tumor suppression is a classical strategy for cancer treatment. Bortezomib, a prototype PI drug, is effective against malignancies with high UPS dependence, such as multiple myeloma (MM) and mantle cell lymphoma. To date, three PI drugs are available on the market: Bortexomib (a first-in-class PI), Carfilzomib (a second-in-class PI), and Ixazomib (a first oral PI). Second-generation PIs have been clinically developed, including the oral active agents Oprozomib, Delanzomib, and Marizomib, which have different pharmacokinetic properties. Phase I/II trials demonstrated the efficacy of Oprozomib monotherapy in patients with MM, but have raised pharmacokinetic profile and safety issues that need improvement [22]. The feasibility of Delanzomib has been demonstrated in early-stage clinical studies of MM and solid tumor [23]. However, clinical development of Delanzomib for MM therapy has been halted because of its significant toxicity [24]. Marizomib is more lipophilic and less neurotoxic drugs. In phase III trial, Marizomib is being assessed for the treatment of malignant glioblastoma in combination with temozolomide and radiotherapy (NCT03345095). MM has been studied as a top priority target for PI drugs because of its ability to produce large amounts of IgG from plasma cells. The high protein turnover in myeloma cells and the preferential susceptibility of malignant cells compared to normal cells result in a favorable therapeutic window for PIs in this disease [25]. The main challenges associated with PI drugs are acquired resistance and low efficacy due to poor pharmacokinetic/pharmacodynamic profiles.

E1 inhibitors: Inhibitors of the ubiquitination initiator E1 enzymes comprise a new class of cancer therapeutics [26]. Compared with PIs, E1 inhibitors are expected to induce broader and more effective biological effects, because they interfere with proteasome degradation and Ub-dependent signaling pathways, such as DNA repair and NF-κB signaling. The success of MLN4924 (Pevonedistat), an inhibitor of the E1 enzyme responsible for Neddylation in cancer [27], has led to the development of other E1 enzyme inhibitors involved in Ub/Ubl modification. MLN4924 is currently being studied in phase III trials for patients with leukemia (NCT04090736, NCT03268954). Among them, TAK-243 (MLN7243) is a first-in-class UAE (ubiquitin-activating enzyme) inhibitor. This agent potently inhibits two key Ub E1 enzymes UBA1 and UBA6 in vitro and is effective in preclinical models of solid and hematological tumors [28]. The TAK-243 inhibitor has been registered in a phase I clinical trial in patients with leukemia (NCT03816319). The research is scheduled to begin this year (2022). Another phase I clinical trial of TAK-243 for advanced solid tumors (NCT02045095) was also conducted, but was terminated for sponsorship reasons.

E3 ligase inhibitors: Because E3 ligase is an important factor in determining substrate specificity in ubiquitination, cancer researchers have been interested in regulating this enzyme. For example, MDM2 and HDM2 (the human counterpart of MDM2) have been extensively validated as potential anticancer drug targets [29]. Several MDM2 inhibitors, including Milademetan (NCT05012397, NCT04979442), APG115 (NCT03781986, NCT04358393), Idasanutlin (NCT04029688, NCT02633059), AMG232 (NCT04190550, NCT03031730), BI-907828 (NCT03449381, NCT05218499), and Siremadlin (NCT05155709, NCT05180695), are currently under clinical investigation for cancer. However, features of E3, such as the lack of a canonically active site for E3 ligase, extensive protein-protein interactions with other proteins, and a multi-domain structure, limit the development of high-potency inhibitors. Protein-targeted chimeras (PROTAC), which chemically link proteins with E3 to induce targeted degradation through the UPS, is a new technology that generates proteolytic agents. PROTAC has drawn considerable attention for research in recent years and is expected to be developed as a cancer treatment in the future [30].

DUB inhibitors: The activation (or inhibition of degradation) of DUB, which acts as a tumor suppressor, as well as the inhibition of DUB, which acts as an oncoprotein, may also be promising cancer treatment strategies [31]. Efforts to develop chemical probes and drugs have reported more than 50 DUB inhibitors, and significant advances have been made in the development of tools for biochemical analysis of DUB [32]. However, only a few DUB inhibitors, such as VLX1570, have entered clinical trials (phase I), but were terminated prematurely due to severe toxicity [33]. Mitoxantrone, a US Food and Drug Administration (FDA)-approved drug that can inhibit DUB ubiquitin-specific peptidase 11 (NCT02724163, NCT05313958), and potential DUB ubiquitin-specific peptidase 14 inhibitors, such as 6-mercaptopurine (NCT00866918, NCT00482833) and 6-thioguanine (NCT05276284, NCT00549848), are currently in clinical trials to treat several types of cancers, including leukemia.

The UPS-targeted drugs under clinical investigation for the treatment of cancer are summarized in Table 2.

#### 2.1.3. Abnormal Ub Pathway in Neurodegenerative Diseases

Many neurodegenerative disorders (ND), such as Alzheimer’s disease (AD), Parkinson’s disease (PD), amyotrophic lateral sclerosis (ALS), and Huntington’s disease (HD), present distinct clinical symptoms depending on the localization of the brain pathology. However, they exhibit common neuropathological characteristics, such as the accumulation of misfolded or aggregated proteins. Abnormalities in the Ub-dependent proteolytic pathways have been associated with neurotoxic aggregate formation and consequent neurodegeneration [37]. There is growing evidence that UPS-associated enzymes play a role in neurodegenerative mechanisms, including the aggregation and accumulation of proteins, autophagy, oxidative stress, apoptosis, and aberrant glutamine transduction. The altered expression or presence of mutations in the UPS components reported in patients with ND are summarized in Table 3.

Alzheimer’s disease (AD): UPS dysfunction is caused by Ub mutations associated with AD and other tauopathies. One such protein is a transcriptional frameshift mutant form of Ub (Ubb^+1^), in which the C-terminal Gly76 is replaced by a tyrosine with a 20-residue extension [46]. Ubb^+1^ has been found to specifically accumulate in neurofibrillary tangles and neuritic plaques in the brain tissues of patients with AD [46]. Lack of C-terminal Gly76 renders Ubb^+1^ unable to ubiquitinate other proteins and instead terminates poly Ub chains, rendering it resistant to DUB [47]. Additionally, the overexpression of Ubb^+1^ triggers mitochondrial impairment and cell death in neurons, suggesting that Ubb^+1^ contributes to AD progression [48]. The importance of Ub E3 ligases in AD pathogenesis has been increasingly recognized. For example, Nedd4-1 (neural precursor cell expressed developmentally downregulated protein 4-1), a HECT (Homologous to the E6-AP Carboxyl Terminus) family of E3 ligases, targets several ND-related proteins, including the insulin/insulin growth factor 1 receptor (IGF-1R) [49]. In AD patients, IGF-1 signaling is significantly impaired, and the overall expression of IGF-1R is downregulated [50]. More recent studies have demonstrated that Nedd4-1 promotes the ubiquitination of alpha-amino-3-hydroxy-5-methyl-4-isoxazoleproprionic acid receptor (AMPAR), the primary mediator of synaptic transmission, in response to amyloid β (Aβ). This process triggers AMPAR internalization, accompanied by loss-of-surface AMPAR, resulting in synaptic weakening [51]. The reduction in AMPAR protein levels is one of the molecular features of the early stages of AD [52]. Upregulation of Nedd4-1 has been reported in human AD, PD, and HD brains, and in spinal cords of patients with ALS [41]. There have been no reports of pathogenic mutations at the genetic level. Another HACT domain E3 ligase, UBE3A, plays a key role in synaptic function [53]. Loss-of-function mutations in the UBE3A gene cause neurodevelopmental disorders, such as Angelman syndrome [54]. Unsurprisingly, downregulation of UBE3A has been found in a mouse model of AD, Tg2576. UBE3A deficiency is associated with Aβ metabolism and synaptotoxicity in mice [53,55].

Parkinson’s disease (PD): Parkin, a ring-to-ring-type E3, is the first E3 enzyme targeting α-synuclein [56]. Mutations in the gene encoding parkin are now thought to account for approximately 50% of early-onset PD cases and 10–20% of adolescent PD cases [57]. All known mutations in parkin are loss-of-function mutations. Parkin exerts neuroprotective effects by inducing mitophagy, together with PTEN-induced putative phosphatase 1 (PINK1), to clear damaged mitochondria. In addition, the E3 activity of parkin is involved in promoting the expression of the NF-κB pathway, providing further evidence for the neuroprotective role of parkin [58]. Loss of parkin function promotes early-onset PD and potential cancer progression. In addition to parkin, ubiquitin C-terminal hydrolase L1 (UCHL1) is associated with familial PD. For example, the I93M missense mutation in UCHL1 has been linked to a rare autosomal dominant form of familial PD. For example, the I93M missense mutation in the UCHL1 has been linked to a rare, autosomal dominant form of familial PD [59]. Transgenic mice overexpressing the I93M substitution showed significantly induced dopaminergic neuronal loss [60]. Furthermore, the downregulation and extensive oxidation of the UCHL1 protein have been found in idiopathic PD brains [61]. In this study, reduced UCHL1 was also found in the cortex tissues of patients with AD, suggesting a pathological feature common to both AD and PD. Many E3 enzymes are able to ubiquitinate α-synuclein. Among them, Nedd4-1 potently promotes α-synuclein ubiquitination and clearance in cells [62] and in vivo [63]. Moreover, enhanced Nedd4-1 activity indicates a neuroprotective function against α-synuclein-induced toxicity in two animal models of PD, fruit flies and rats [63].

#### 2.1.4. UPS Inhibitors as Treatment for Neurodegenerative Diseases

As a potential treatment strategy for ND, the Ub pathway is also being studied to enhance ubiquitination (e.g., maintaining Ub retention or upregulating E3 ligase activity), enhance proteasome activity, and inhibit polyubiquitin chain trimming. As with cancer, small molecules are currently being developed by targeting different mechanisms, such as the inhibition or activation of specific E3 ligases or DUBs, and the correction of possible misregulations in the UPS. However, most drugs targeting the Ub pathway, including DUB inhibitors used as ND therapies, are still in preclinical development (Table 4).

USP14 inhibitors: The inhibition of Ub-specific protease 14 (USP14), a proteasome-associated DUB enzyme that binds to proteasomes, is a strategy to promote efficient proteasomal clearance of ubiquitylated proteins by preventing premature trimming of the Ub chain on the substrate [64]. In cultured cells, selective USP14 inhibitors, such as IU1, promote proteasome-mediated degradation of ND-associated proteins, such as Tau, TAR DNA binding protein 43 (TDP-43), and ataxin-3 [64,65]. In a PD fruit fly model, genetic or pharmacological inhibition of USP14 improved the phenotype of the disease and prolonged lifespan of the organism [66]. However, IU1 treatment reduces autophagy activity in cultured cells and induces the accumulation of HTT proteins with long, pathogenic polyQ repeats [67] showing a preference for autophagy-mediated clearance. Although USP14 inhibition may be substrate-specific, a reduction in the autophagy-lysosome pathway may be a potential neurotoxic mechanism. Further animal model studies are needed to validate the clinical usefulness of USP14 inhibition in the treatment of ND.

UCHL1 inhibitor: Another important DUB enzyme associated with ND pathogenesis is ubiquitin carboxyl-terminal hydrolase isozyme L1 (UCHL1), highly specific to neuron and is one of the most abundant soluble proteins in the brain [72]. UCHL1 binds to the PD-relevant protein α-synuclein, and pharmacological inhibition of UCHL1 in oligodendroglia prevents aggregation of α-synuclein by promoting autophagy [69]. Reducing UCHL1 activity, either by LDN57444 or genetic deletion, alleviates PD-related phenotypes, including reduced climbing ability and loss of dopaminergic neurons, by regulating glucose metabolism in PINK1- or Parkin-deficient mutant flies [70]. To date, LDN57444 is the only reported UCHL1 inhibitor that exhibits chemical instability and off-target toxicity [73]. Over the past few years, Mission Therapeutics, a drug development company targeting UPS in the UK, has developed cyanopyrrolidine derivatives as new inhibitors of UCHL1 and has filed a patent for the results [68]. However, information regarding their inhibitory potency in cells and their biological activities is still very limited.

USP30 inhibitors: The inhibition of Ub carboxyl-terminus hydrolase 30 (USP30), which acts as an antagonist of parkin, is a new strategy to ameliorate mitophagy deficits in PD. The mitochondrial-localized DUB USP30 can selectively induce mitochondrial ubiquitination, making it a particularly attractive drug target. Neuronal USP30 suppression improves mitophagy defects, motor function, and organism survival induced by parkin mutants in the *Drosophila* PD model [74]. Pharmaceutical companies are working to identify USP30 inhibitors, including phenylalanine derivatives, cyano-pyrrolidines, and phenyl- or naphthyl sulfonamide derivatives. For example, several small-molecule USP30 inhibitors have been developed by Mission Therapeutics and Mitobridge (Astellas Pharma Company), including MTX652, MTX114, and MF0094 [71]. Animal studies, including USP30 knockout studies, are essential for the future development of these therapeutics.

Pimozide: Pimozide has been approved by the US FDA as an antipsychotic drug. This compound was initially screened out as an anticancer drug, targeting its reversible inhibitory activity against the USP1/USP1-associated factor 1 (UAF1) complex [75]. Pimozide re-sensitizes NSCLC resistant to cisplatin, a platinum-based chemotherapeutic drug with sub-micromolar potency [75]. One substrate of the USP1/UAF1 deubiquitinase complex is TANK-binding kinase 1 (TBK1), a key regulator of autophagy [76]. In ND, heterozygous loss-of-function mutations in TBK1 have been identified as the cause of ALS [77]. USP1/UAF1 inhibition may promote TBK1 stabilization. Therefore, Pimozide is currently undergoing a phase II clinical trial in ALS patients in Canada (NCT03272503). Short-term (6 weeks) administration of this drug stabilizes the right hand muscles in patients with ALS [78].

### 2.2. SUMOylation

Small ubiquitin-related modifiers (SUMOs) are the most extensively studied UBLs. Similar to ubiquitination, SUMOylation/deSUMOylation requires E1-E2-E3 enzymes and isopeptidases [79]. These enzymes are SUMO-specific and do not overlap with enzymes used for ubiquitination. UBC9, the sole SUMO-specific E2 enzyme in mammals, can directly transfer SUMO to hundreds of SUMO substrates by directly binding to the SUMO consensus motif ψ-K-X-E, where ψ is a hydrophobic amino acid, and X can be located in any amino acid. Covalent binding of SUMO to a substrate often modulates its target function by recruiting other cellular proteins. Most known SUMO E3 enzymes harbor one or more SUMO interaction/binding motifs that mediate non-covalent interactions with SUMO. SUMOylation is a key regulator of gene expression, chromatin remodeling, ion channel activity, signal transduction, and sensing of oxidative stress, as it can alter the localization, activity, protein-protein interactions, and stability of its substrates [79]. The SUMOylation process and drugs targeting the different steps are shown in Figure 3.

#### 2.2.1. Dysregulation of the SUMO Pathway in Cancer

In cancer cells, SUMO modifications are involved in carcinogenesis, differentiation, proliferation, metastasis, and apoptosis by regulating the DNA damage response, cell cycle, and cell-to-cell communication. PTMs can contribute to oncogenic or tumor suppressor signaling pathways in a context-dependent manner.

Abnormalities in the SUMOylation machinery in cancer tissues have been detected at the genetic, epigenetic, mRNA, and protein levels. Large-scale data from The Cancer Genome Atlas (TCGA) show that most SUMOylation/deSUMOylation-associated enzymes are upregulated in cancer, indicating an increased utility of SUMO modification in cancer cells. In some cases, their levels are positively correlated with the cancer stage, malignancy, and poor prognosis. For example, unbiased genetic screens have identified SAE1/SAE2, subunits of the SUMO-activating enzyme E1, as a genetic driver of Myc-dependent tumorigenesis [80]. The upregulation of SAE1/SAE2 is correlated with lower survival rates in patients with Myc-high breast cancer [80]. In addition, hyper-SUMOylation, which is associated with SAE1/SAE2 upregulation, is required for the progression of Myc-induced hematological malignancies [81] and small-cell lung cancer with high Myc expression [82]. SNPs in the UBC9 (UBE2I) and PIAS3 genes are associated with higher breast cancer risk and tumor grade [83]. In patients with MM, both gene and protein levels of UBC9 and PIAS1 are elevated and negatively correlated with patient survival [84]. Genetic mutations and the abnormal expression of SUMO deconjugases SENPs have also been reported. According to TCGA data, SENP isoforms are upregulated in most cancers [85]. A genome-wide epigenetic examination of clinical samples revealed promoter hypomethylation of SENP6 in hepatocellular carcinoma (HCC), providing evidence that elevated levels of SENP6 are associated with the promotion of HCC tumorigenesis [86]. The types of cancer for which SUMOylation defects have been reported in patients are summarized in Table 5.

#### 2.2.2. SUMO Pathway Inhibitors Used to Treat Cancer

Several SUMOylation inhibitors with anticancer potential have been reported [92]. Examples of SUMOylation inhibitors are listed in Table 6 and Figure 3.

SUMO E1 (SAE) inhibitors: Several natural compounds, including ginkgolic acid [93], davidiin [94], tannic acid [95], and kerriamycin B [96], have been shown to inhibit SUMO E1. However, owing to the multifaceted pleiotropic effects of natural product-oriented SUMOylation inhibitors, much attention is being paid to the development of synthetic inhibitors with higher SUMO E1 complex specificity. For example, ML-792, a nanomolar potency small-molecule inhibitor of SUMO E1, has been developed. This compound forms a conjugate with SUMO, subsequently binds to SAE2, the catalytic subunit of SUMO E1, and inhibits its activity [97]. The selective cytotoxicity of ML-792 is elevated in cells overexpressing c-Myc [97]. TAK-981 is a mechanism-based inhibitor derived from ML-792, designed to improve the retention period of the TAK-981-SUMO adduct in vivo [98]. This agent inhibits SUMOylation (i.e., loss of SUMO2/3-protein conjugates) for approximately 15 to 20 h in human lymphoma xenograft mice treated with 10 mg/kg [99]. To date, TAK-981 (Subasumstat) is the only compound that directly targets SUMO E1 and SUMOylation, and has successfully entered clinical investigations in cancer patients. In vivo preclinical studies demonstrate TAK-81′s synergistic effects in combination with other immunomodulatory agents for cancer treatment [99]. Phase I/II studies are ongoing for the treatment of patients with solid tumors and lymphomas (monotherapy: NCT03648372; in combination with immunotherapy drugs: NCT04065555, NCT04074330, NCT04381650, and NCT04776018).

UBC9 inhibitors: UBC9 is the only conjugating enzyme in the SUMOylation process and is upregulated in various types of tumors [100], making it an attractive drug target. However, as in the case of Ub E2, efficient drug design to inhibit UBC9 is difficult due to its rigid structure, the absence of drug-processable pockets, and the presence of many protein-protein interaction sites. Several UBC9 inhibitors, such as 2-D08 [101] and spectomycin B1 (natural product) [102], show some promise in vitro because of their anticancer effects. For example, 2-D08 inhibits the migration of pancreatic cancer cells by regulating the oncogenic or pathogenic activities of K-Ras [103]. 2-D08 inhibitor promotes excessive ROS-mediated intrinsic mitochondrial apoptosis of acute myeloid leukemia (AML) cells [104]. Recent studies have indicated the potential benefits of 2-D08 inhibitors in the treatment of cancer combined with chemotherapy. Co-treatment with 2-D08 and etoposide, a clinically used chemotherapeutic drug, enhances the sensitivity of tumor cells to etoposide in vivo [105]. In many cases, despite the profound efficacy of 2-D08 found in vitro, its antitumor activity in vivo is unclear, which may be due to its low solubility and bioavailability [106]. Spectomycin B1 also validated its ability to bind directly to UBC9 and inhibit the formation of the UBC9-SUMO complex in the low-micromolar range. Antagonizing UBC9 by spectomycin B1 treatment prevents estrogen-dependent proliferation in MCF7 human breast cancer cells [102]. These results suggest the potential antitumor activity of spectomycin B1 against hormone-dependent breast cancer. However, these two agents have not been used in clinical studies. The development of other UBC9 chemical inhibitors using a small-molecule microarray approach is underway [107].

SENP inhibitors: The development of SENP-directed inhibitors began relatively early on. Although isoform inhibitors can improve drug efficacy and safety compared to pan-SENP inhibitors, the design of isotype-selective inhibitors is difficult because of their isopeptidase cleavage chemistry, protein structure, and the presence of similar amino acid sequences within the catalytic site [85]. Currently, SENP inhibitors mainly target SENP1, which is the most commonly analyzed inhibitor in clinical samples, and is a therapeutic target in ex vivo clinical studies (NCT03798587 and NCT04167605). SENP1 inhibitors have specifically been reported for the treatment of prostate cancer. For example, GN6958 [108] and SI2 [109] show SENP1-specific inhibitory activity in the micromolar range and exhibit antitumor activity in prostate cancer cells. Other reported SENP1 inhibitors are triterpenoids [110] and monomeric Ic [111]. These two natural products have demonstrated preclinical efficacy in vitro and in prostate cancer xenograft models. Approximately ten more peptides or compounds have been reported as SENP inhibitors without further efficacy studies. No SENP inhibitors have been clinically studied.

**Table 6 ijms-23-05053-t006:** Selected inhibitors target in SUMO pathway.

	Compound ID	Description	Stage
E1	TAK-981 (Subasumstat)	The first-in-Class SAE Inhibitor	Phase I/II trials (https://clinicaltrials.gov/ct2/show/NCT03648372, accessed on 1 March 2022)
	Ginkgolic acid, davidiin, tannic acid, kerriamycin B, Pyrazole, and thiazole urea containing Cpds	SAE inhibitor	Preclinical [93,94,95,96,112]
E2	2-D08, Spectomyin B	UBC9 inhibitor	Preclinical [101,102,103,104,105,106]
SENP	GN6958, Triterpenoids, Monomeric Ic	SENP1 inhibitor	Preclinical [108,110,111]
	JCP666 and its analogues	SENP1/2 inhibitor	Preclinical [113]
	SUMO-1-VS, Ebselen and 6-thioguanine	SENP2 inhibitor	Preclinical [114,115]
	SI2	SENP1/2/3 inhibitor	Preclinical [109]
	SPI-01	SENP1/2/7 inhibitor	Preclinical [109]
	VEA499/VEA561	SENP1/2/6/7 inhibitor	Preclinical [113]

Abbreviations: Cpd, compounds; SAE, SUMO-activating enzyme; UBC9, ubiquitin-conjugating enzyme 9; SENP, SUMO-specific proteases.

#### 2.2.3. Role of SUMOylation in Neurodegenerative Disease Pathogenesis

Dysregulation of SUMOylation is likely involved in ND, because SUMO targets multiple proteins involved in neural development and function [116]. Studies have shown that protein SUMOylation directly regulates neurogenesis and plays a role in recovery from brain damage caused by cerebral ischemia, a causative factor of ND. Cerebral ischemia in mice and humans results in significant increases in SUMOylated proteins in the region surrounding cerebral lesions [117,118]. This PTM also plays a protective role against cellular stresses, such as glucose and oxygen starvation in neurons [119]. It is important to note that SUMO frequently co-localizes with neural inclusions, and SUMO modifications can enhance or prevent the formation of pathogenic protein aggregates in many cases of ND (Table 7). In addition, genetic mutations and altered expression of the SUMO machinery have been found in patients with ND. For example, an analysis of the genomic DNA of patients with late-onset AD revealed that polymorphisms in the UBC9 gene are significantly associated with the disease [120]. The downregulation of SENP3 mRNA expression [121] and SUMO2 levels [122] in the brain tissues of patients with AD is another example of the relationship between abnormal SUMOylation and AD pathological mechanisms.

#### 2.2.4. Therapeutics Targeting the SUMO Pathway in Neurodegenerative Diseases

Despite the importance of SUMOylation in neurodegeneration, functional studies have yielded inconsistent results, and appropriate animal models for studying SUMO in many NDs have not yet been established. Nevertheless, given the importance of proteolysis, SUMOylation inhibitors are gaining attention in ND therapy, and recent studies have suggested their potential as therapeutics. For example, ginkgolic acid, a natural compound with SUMO E1 inhibitory activity, reduces the levels of the mutant form of huntingtin and activates autophagy flux in cultured HD striatal cells [129]. 2-D08, a specific inhibitor of UBC9, exhibits concentration-dependent protective efficacy over the range between 10 to 100 µM against amyloid β protein neurotoxicity in cultured cells [130]. Computational modeling further showed that 2-D08 could bind to the 42 amino acid form of Aβ_1–42_ monomer with high affinity, thereby eliminating the pathological form of amyloid β.

There have also been reports of drug-like small molecules that enhance SUMOylation in ND. Krajnak and Dahl identified three small-molecule SUMOylation activators that directly targeted SUMO E1, and confirmed their neuroprotective efficacy in cultured cells. [131]. Given the findings of a study on cerebral ischemia, a temporary increase in SUMOylation at the border of the ischemic tissue is expected to have a therapeutic effect [132]. Studies using mouse neural stem cells (NSCs) have provided further evidence that upregulated SUMOylation is an essential component of ischemic tolerance. Activation of SUMOylation through UBC9 overexpression in NSCs confers a higher predisposition to differentiate into neurons in the brains of stroke mice. Further studies are needed to investigate whether pharmacological manipulation of SUMOylation can improve the outcomes of NSC-based therapy [133]. Currently, no SUMO-specific drugs are in clinical trials for patients with ND.

#### 2.2.5. Dysregulation of SUMOylation in Heart Disease

SUMO modification is essential for the development, functioning, and pathogenesis of heart disease. In murine models, reduced SUMOylation activity, such as SUMO1 deficiency or SENP2 overexpression, can lead to fatal congenital heart disease [134,135]. SUMO1 knockdown induces contractile dysfunction and structural abnormalities in mice [136], whereas mouse hearts overexpressing SUMO2 or SENP5 develop extensive cardiac muscle dysfunction [137,138]. Similar to the other diseases described above, abnormalities in the SUMO machinery components have been found in humans, including heart failure, cardiomyopathy, and myocardial infarction (Table 8). There are no reports of genetic mutations in the SUMOylation component associated with heart disease. Similar to those in other organs, critical cardiac proteins, including sarcoplasmic reticulum Ca^2+^-ATPase pump (SERCA2a), responsible for stress adaptation and cardiac function, are physically and functionally linked to SUMOylation [139,140].

#### 2.2.6. Therapeutics Targeting the SUMO Pathway

Although clinical studies involving drugs targeting this pathway are not yet underway, SUMOylation has recently emerged as an exciting PTM that positively contributes to cardiac function and protein maintenance. For example, SUMO enhances the activity and protein stability of SERCA2a, a heart failure disease-modifying protein [145]. In murine and porcine models of heart failure, recovery of SUMO1 expression via gene transfer significantly improved cardiac contractility and hemodynamic parameters by restoring SERCA2a activity and SUMOylation. SUMO1 overexpression and gene transfer also restored SERCA2a downregulation and SERCA2a SUMOylation, resulting in significantly improved cardiac contractility and hemodynamic parameters [134,146]. More importantly, mouse studies suggest the potential of compound N106, an activator of SERCA2a SUMOylation that targets the SUMO E1 enzyme, as a therapy for heart failure [147]. Additionally, the cardioprotective effect of luteolin, a flavonoid with anticancer potential, has been reported to occur through SERCA2a SUMOylation [148]. These studies indicate that SUMOylation is a promising target for heart failure therapy.

An interesting study has suggested the importance of SUMOylation in cardiac protein quality control. Increased UBC9 expression enhances proteasome function and reduces protein aggregate levels in cell cultures and murine models of protein toxicity [149]. Given the results of ND study, targeting the ability of SUMOylation to regulate toxic proteins and interact with the UPS will be another novel therapeutic strategy. The identification and functional studies of SUMO-targeted ubiquitin ligases and SUMO deubiquitinases are currently ongoing [150].

## 3. Conclusions

Ub and Ubl are essential molecules for cellular homeostasis and function, and their modification processes are often poorly regulated in diseased states. Recently, ubiquitination and SUMOylation have attracted attention as novel therapeutic targets for complex diseases, such as cancer, neurodegenerative disorders, and heart disease. Many clinical investigations are underway on Ub- and SUMO-specific drugs. Synergistic effects of these drugs can be expected in clinical settings, because ubiquitination and SUMOylation not only modify the target protein, but are also associated with communication with other major PTMs, including phosphorylation and acetylation. Further studies on the disease-specific substrates of Ub/Ubl, the detailed molecular basis of modulating their modification process, and their physical and functional relationships with other PTMs will increase the likelihood of developing Ub/Ubl modulators as next-generation treatments for complex human diseases.

## Figures and Tables

**Figure 1 ijms-23-05053-f001:**
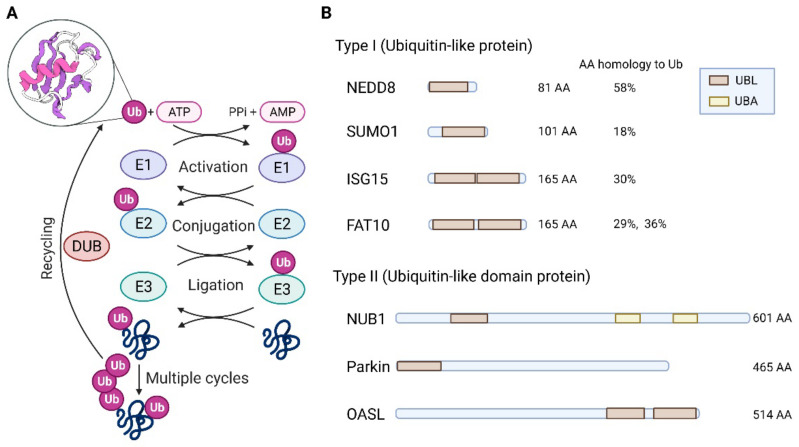
Biochemical and Structural Features of Ubiquitin & Ubiquitin-like Proteins. (**A**) Main steps in the substrate modification process by Ub. The β-grasp globular fold structure of Ub is shown in the circle. (**B**) Examples of structural domains within human type I and type II Ub family members. DUB, de-ubiquitinating enzymes; FAT10, HLA-F-adjacent transcript 10; ISG15, Interferon-stimulated gene 15; NEDD8, Neural precursor cell-expressed developmentally down-regulated gene 8; NUB1, NEDD8 ultimate buster 1; OASL, 2′-5′-oligoadenylate synthetase-like protein; SUMO1, small ubiquitin-like modifier 1; Ub, ubiquitin; UBL, ubiquitin-like domain; UBA, ubiquitin-associated domain. Figure 1A adapted from “ubiquitination”, by BioRender.com (accessed on 1 March 2022). Retrieved from https://app.biorender.com/biorender-templates. Agreement number is OU23V77K75.

**Figure 2 ijms-23-05053-f002:**
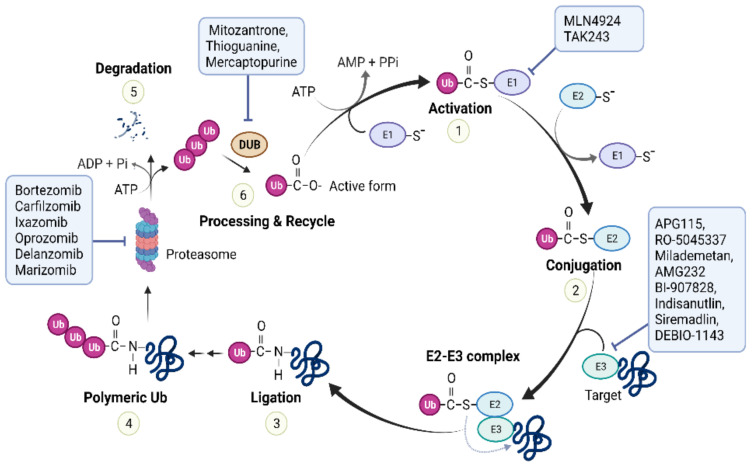
The ubiquitin-proteasome system (UPS). In step 1, ubiquitin is activated by a ubiquitin activating enzyme (E1). In step 2, activated ubiquitin (Ub) is transferred to a ubiquitin conjugating enzyme (E2). In step 3, ubiquitin is transferred from E2 to a specific target (substrate) lysine to form a covalent bond. This process is catalyzed by E3 ubiquitin ligase. In step 4, repeated ubiquitin conjugation generates a poly-ubiquitin chain. In step 5, the ubiquitinated target (e.g., canonical K48 linkage) is recognized, unfolded, and digested by the 26S proteasome. In step 6, Deubiquitinating (DUB) enzymes remove covalently linked ubiquitin moieties from ubiquitin-ubiquitin and ubiquitin-protein conjugates. These enzymes also process ubiquitin precursors to generate free (un-anchored) ubiquitin pools. The free ubiquitin can be further reused. Inhibitors of various UPS components are being developed clinically for the treatment of cancer and neurodegenerative diseases.

**Figure 3 ijms-23-05053-f003:**
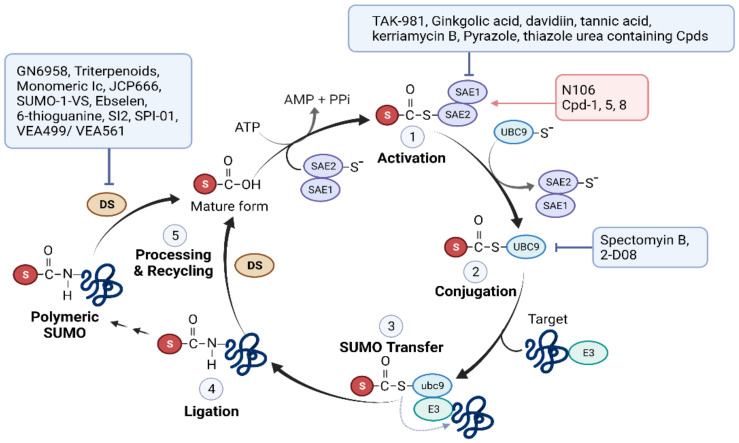
Small molecules targeting the SUMO pathway. In step 1: SUMO E1 (SAE1/SAE2 heterodimer) activates mature SUMO through an ATP-dependent reaction. SAE1 catalyzes adenylation of the C-terminus of SUMO to form a SUMO-AMP intermediate, which then transfers SUMO to SAE2 and forms a thioester bond. In step 2: SUMO is transferred from SAE2 to SUMO E2 (UBC9), which in turn forms a thioester bond. In steps 3 and 4: UBC9 catalyzes the formation of an isopeptide bond between the C-terminal glycine of SUMO and a lysine residue on the substrate. Certain SUMO E3 ligases are often involved in this process. In step 5: SUMO is removed from the lysine residue of the target proteins by a de-SUMOylation enzyme (DS), such as SENP, sentrin-specific protease. SENP also catalyzes SUMO maturation by cleaving the precursor SUMO at C-terminus, exposing two glycine residues required for conjugation.

**Table 1 ijms-23-05053-t001:** Examples of cancer-related UPS enzymes. The relevant cancer type information for each gene referenced OMIM (https://www.ncbi.nlm.nih.gov/omim) (accessed on 7 March 2022) or The Cancer Genome Atlas data.

	Gene Name	Function in Cancer	Deregulation	Cancer Type
E1	UBA1	Oncogene	↑	LNC [7], SCC [8], MM [9], PM in LC, TC [TCGA]
E2	UBE2A	Oncogene	Mutation, ↑	HCC [12] CML [13], PM in RCC, CC, HNC [TCGA]
	UBE2C	Proto-oncogene	↑	BC, PC, CRC, OC, Lymphoma [14], PM in RCC, LC, PC [TCGA]
	UBE2D	Oncogene	↑	PM in RCC [TCGA]
	UBE2L	Oncogene	↑	NSCLC [10], PM in BC, OC [TCGA]
	UBE2N	Oncogene	↑	BC, PC, CRC, OC, Lymphoma [14], PM in LC [TCGA]
	UBE2S	Oncogene	↑	PM in RCC, LC, EC [TCGA]
	UBE2T	Oncogene	↑	PM in RCC, LC, OC [TCGA]
E3	APC3 (CDC27)	Tumor suppressor	Mutation, ↑	PC [15], PM in RCC, CRC, LC, LNC [TCGA]
	BRCA1	Tumor suppressor	Mutation	Familial BC, OC [OMIM 113705]
	CBL	Proto-oncogene	Mutation	Leukemia [OMIM 165360]
	FBW7	Tumor suppressor	Mutation	BRC, CRC, EC [16], PM in RCC [TCGA]
	MDM2	Oncogene	↑	PM in EC, CC [TCGA]
	SKP2	Oncogene	↑	PM in RCC, Melanoma, OC [TCGA]
	PAKN2	Tumor suppressor	Mutation	LNC, OC [OMIM 602544]
	VHL	Tumor suppressor	Mutation	RCC [OMIM 608537], PM in LC, BC, SC [TCGA]
DUB	BAP1	Tumor suppressor	Mutation	BC, LC, RCC [OMIM 603089], PM in RCC [TCGA]
	CYLD	Tumor suppressor/Oncogene	Mutation	Familial cylindromatosis, Trichoepithelioma [OMIM 605018]
	FANCL	Tumor suppressor	Mutation	Fanconi leukemia [OMIM 608111], PM in RCC, LC, UC [TCGA]
	TNFAIP3	Tumor suppressor/Oncogene	Mutation, ↑	BC, Lymphomas [OMIM 191163], PM in RCC [TCGA]
	USP4	Oncogene	Mutation, ↑	LNC [OMIM 603486], PM in RCC, LNC [TCGA]
	USP7	Tumor suppressor/Oncogene	↑	OC [17], Glioma [18], PM in RCC [TCGA]
	USP14	Oncogene	↑	PM in LC, HNC, OC [TCGA]
	USP28	Oncogene	↑	BC [19], CRC [20]
	USP42	Oncogene	↑	GC [21]

Abbreviations: BC, breast cancer; BRC, bladder cancer; CC, cervical cancer; CML, chronic myelogenous leukemia; CRC, colorectal cancer; EC, endometrial cancer; GC, gastric cancer; HCC, hepatocellular carcinoma; HNC, head and neck cancer; LC, liver cancer; LNC, Lung cancer; NSCLC, Non-small cell lung cancer; MM, multiple melanoma; OC, ovarian cancer; PC, pancreatic cancer; RCC, renal cell carcinoma; SC, stomach cancer; SCC, squamous cell carcinoma; TC, thyroid cancer; UC, urothelial cancer; ↑, Upregulation; PM, Prognostic marker; TCGA, The Cancer Genome Atlas.

**Table 2 ijms-23-05053-t002:** List of anticancer drugs targeting UPS. Drugs under FDA/EMA approval or clinical investigation are summarized. For more detailed clinical information, see ClinicalTrials.gov. All accessed on 1 March 2022.

	Name	Description	Highest Stage	References
E1	MLN-4924 (Pevonedistat)	The first-in-class NAE/UAE inhibitor	Phase III	https://clinicaltrials.gov/ct2/show/NCT04090736
	TAK-243 (MLN-7243)	The first-in-class UAE/E1 inhibitor	Phase I	https://clinicaltrials.gov/ct2/show/NCT03816
E3	RO-5045337	The first-in class MDM2 inhibitor	Phase I	https://clinicaltrials.gov/ct2/show/NCT0062387
	Milademetan APG115 Idasanutlin AMG232 BI-907828 Siremadlin	p53-MDM2 inhibitor	Phase I/II	https://clinicaltrials.gov/ct2/show/NCT04029688 https://clinicaltrials.gov/ct2/show/NCT02633059 https://clinicaltrials.gov/ct2/show/NCT04358393 https://clinicaltrials.gov/ct2/show/NCT05012397 https://clinicaltrials.gov/ct2/show/NCT04979442 https://clinicaltrials.gov/ct2/show/NCT01985191 https://clinicaltrials.gov/ct2/show/NCT04190550 https://clinicaltrials.gov/ct2/show/NCT03031730
	DEBIO-1143	IAP inhibitor	Phase III	https://clinicaltrials.gov/ct2/show/NCT04459715
DUB	Mitozantrone	USP11 inhibitor	Phase III	https://clinicaltrials.gov/ct2/show/NCT02724163 https://clinicaltrials.gov/ct2/show/NCT05313958
	Mercaptopurine Thioguanine	USP14 inhibitor	Phase III	https://clinicaltrials.gov/ct2/show/NCT00866918 https://clinicaltrials.gov/ct2/show/NCT00482833 https://clinicaltrials.gov/ct2/show/NCT05276284 https://clinicaltrials.gov/ct2/show/NCT03117751
Proteasome	Bortezomib	The first-in class MDM2 inhibitor	FDA approved	[34]
	Carfilzomib	Second-in-class PI	FDA approved	[35]
	Ixazomib	The first oral PI	FDA approved	[36]
	Oprozomib	Oral PI	Phase I	https://clinicaltrials.gov/ct2/show/NCT02939183
	Delanzomib	Oral PI	Phase I/II	https://clinicaltrials.gov/ct2/show/NCT01348919
	Marizomib	Oral PI	Phase III	https://clinicaltrials.gov/ct2/show/NCT03345095

Abbreviations: DUB, de-ubiquitinating enzymes; E1, ubiquitin-activating enzyme; E2, ubiquitin-conjugating enzyme; E3, ubiquitin ligase; FDA, The United States Food and Drug Administration; PI, proteasome inhibitor.

**Table 3 ijms-23-05053-t003:** Examples of neurodegenerative disease-related UPS components. The disease information for each gene referenced OMIM (https://www.ncbi.nlm.nih.gov/omim). (All accessed on 1 March 2022).

	Gene Name	Deregulation Type	Disease
Ub precursor	UBB	Missreading, misframed mutations	AD [OMIM 191339]
E3	CHIP	↑	AD [38]
	FBXO7	Loss-of-function mutations	PD [OMIM 605648]
	HACE1	↓	HD [39]
	HRD1	↓	AD [40]
	LRSAM1	Loss-of-function mutations	PD [OMIM 610933]
	NEDD4-1	↑	AD, PD, ALS [41]
	PRKN (PARK2)	Loss-of-function mutations	PD [OMIM 602544]
	RNF182	↑	AD [42]
	TRAF6	↑	PD [43], HD [44]
	UBE3A	Loss-of-function mutations	AS [OMIM 601623]
DUB	UCHL1 (PARK5)	↓, Loss-of-function mutations	AD, PD [OMIM 191342]
	USP13	↑	PD [45]

**Abbreviations:** AD, Alzheimer’s disease; AS, angelman syndrome; ALS, amyotrophic lateral sclerosis; DUB, de-ubiquitinating enzymes; E3, ubiquitin ligase; HD, Huntington’s disease; PD, Parkinson’s disease; ↑, Upregulation; ↓, Downregulation.

**Table 4 ijms-23-05053-t004:** Selected DUB small molecule inhibitors reported in neurodegenerative disorders.

Compound ID	Description	Stage
Pimozide	USP1 inhibitor	Phase II trials (https://clinicaltrials.gov/ct2/show/NCT03272503, accessed on 1 March 2022)
IU1, IU1 analogs and derivatives	USP14 inhibitor	Preclinical [64,65,66,67]
Cyanopyrrolidine derivatives, LDN57444	UCHL1 inhibitor	Preclinical [68,69,70]
MTX652, MTX114, MF0094	USP30 inhibitors	Preclinical [71]

Abbreviations: USP, ubiquitin-specific peptidase; UCHL1, ubiquitin carboxyl-terminal hydrolase L1.

**Table 5 ijms-23-05053-t005:** Examples of SUMOylation components associated with cancer.

	Gene Name	Deregulation	Caner Type
Modifier	SUMO1	↑	PM in OC [TCGA]
	SUMO2	↑	PM in RC, EC, HCC [TCGA]
	SUMO3	↑	PM in EC [TCGA]
SUMO E1	SAE1	↑	PTC [87] PM in HCC, RC, TC [TCGA]
	SAE2	↑	BC [80], SCLC [81], PM in HCC, RC [TCGA]
SUMO E2	UBC9	↑	Melanoma [88], PM in TC, HCC [TCGA]
SUMO E3	PIAS1	↑	PTC [87], PCA [89]
	PIAS2	↓	PTC [87], PM in TC [TCGA]
	PIAS3	↑ or ↓, Mutation	PM in RC, HCC [TCGA]
	PIAS4	↑	PM in EC, PAC [TCGA]
	RANBP2	↑, Mutation	CRC [90]
deSUMOylase	SENP1	↑	PM in RC, HCC [TCGA]
	SENP2	↓ or ↑, Mutation	PM in EC [TCGA]
	SENP3	↑	PM in PAC [TCGA]
	SENP5	↑	PM in RC, EC, HCC [TCGA]
	SENP6	↑	PM in RC, TC [TCGA]
	SENP7	Long↑; Short varient↓	BC [91]
		↑	PM for HNC [TCGA]

Abbreviations: BC, breast cancer; CRC, colorectal cancer; EC, endometrial cancer; HCC, hepatocellular carcinoma; HNC, Head and neck cancer; PAC, pancreatic cancer; PCA, prostate cancer; RC, renal cancer; TC, Thyroid cancer; SAE, SUMO-activating enzyme; SCC, squamous cell carcinoma; SENP, SUMO-specific proteases; SCLC, small cell lung cancer; OC, ovarian cancer; PIAS, protein inhibitor of activated STAT; PTC, papillary thyroid cancer; UBC9, ubiquitin-conjugating enzyme 9; ↑, Upregulation; ↓, Downregulation. PM, Prognostic marker; TCGA, The Cancer Genome Atlas.

**Table 7 ijms-23-05053-t007:** Neurodegenerative disease-related key protein regulated by SUMO.

Substrate	Substrate’s Function	Functional Impact	Disease
APP	Aβ generation	Negative regulation of Aβ aggregates levels	AD [123]
Tau	Microtubule stabilization	Induction of tau hyper-phosphorylation & inhibition of tau degradation	AD [124]
HTT	Microtubule-mediated transport and vesicle function	Increased cytotoxicity by specifically stabilizing mutant HTT via Rhes	HD [125]
α-Synuclein	PD pathogenesis	Maintanance of α-synuclein in a soluble form	PD [126]
DJ-1	Anti-oxidative stress and transcriptional regulation	Essential for DJ-1 solubility and its function	PD [127]
Parkin	E3 Ub ligase	Induction of Parkin’s self-ubiquitination & nuclear translocation	PD [128]

Abbreviations: Aβ, amyloid-β; AD, Alzheimer’s disease; APP, Amyloid-β precursor protein; HD, Huntington’s disease; HTT, Huntingtin; PD, Parkinson’s disease.

**Table 8 ijms-23-05053-t008:** Heart disease related SUMO components.

	Component	Expression	Regulation Pathway	Disease
Modifier	SUMO1	↓	Heart development, cardiac pathology	CHD, HF [134,136]
	SUMO2/3 conjugates	↑	Cardiac pathology	HF [137]
E2	UBC9	↑	Authophagy	MI, CM [141,142]
Deconjugase	SENP1	↑	Mithocondrial function	HF, MI/R [143,144]
	SENP2	↑	Heart development and function	CHD [134]
	SENP5	↑	Mithocondrial function	HF [138]

Abbreviations: CHD, Congenital heart disease; CM, cardiomyopathy; HF, Heart failure; MI/R, Myocardial Ischemia/Reperfusion injury; MI, Myocardial infarction; ↑, Upregulation; ↓, Downregulation.

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
