# Peer review of "Ubiquitin and Ubiquitin-like Proteins in Cancer, Neurodegenerative Disorders, and Heart Diseases"

_ijms, 2022, doi:10.3390/ijms23095053_

Round 1
Reviewer 1 Report
Authors are encouraged to include 1-2 Figures with the deregulated Ub and SUMO pathways, showing where the currently-used UPS inhibitors and SUMO pathway inhibitors, act. This would help the reader follow these pathways and the mode of action of the different therapeutic molecules.
Author Response
Response to Reveiwers's Comments
We appreciate the reviewers’ comments and revise the manuscript accordingly. The manuscript has been improved by covering all the points of the reviewers. In this revised manuscript, changes are highlighted in red.
Reviewer number 1:
Authors are encouraged to include 1-2 Figures with the deregulated Ub and SUMO pathways, showing where the currently-used UPS inhibitors and SUMO pathway inhibitors, act. This would help the reader follow these pathways and the mode of action of the different therapeutic molecules.
Author response: Thank you for this suggestion. In this revised version, we provided New Figures 2 and 3 showing the relevant drugs targeting the UPS and SUMO pathways, respectively.
Reviewer 2 Report
Hwang Jin-Taek et al summarized the roles of ubiquitin and ubiquitin-like proteins in human diseases. Post translational modification of proteins by ubiquitin or ubiquitin-like proteins plays important roles in regulating the activity of their downstream factors, leading to the alteration of protein localization, stability and activity. As such, the authors reviewed the regulation of these processes and their involvement in human diseases like neurodegenerative disease, heart diseases and cancers. In general, this review brings in a comprehensive review of current knowledge and provides important information for both clinical and basic investigators. In addition, the authors also added in tables to present the data in a more efficient way. However, there are several points the authors need to pay attention to improve the current version.
- The title of Table 1/3 needs to be changed, is "faults" required?
- Should cite more relevant references;
- Be careful about some grammar and typo mistakes.
Author Response
Response to Reveiwers's Comments
We appreciate the reviewers’ comments and revise the manuscript accordingly. The manuscript has been improved by covering all the points of the reviewers. In this revised manuscript, changes are highlighted in red.
Reviewer number 2:
Hwang Jin-Taek et al summarized the roles of ubiquitin and ubiquitin-like proteins in human diseases. Post translational modification of proteins by ubiquitin or ubiquitin-like proteins plays important roles in regulating the activity of their downstream factors, leading to the alteration of protein localization, stability, and activity. As such, the authors reviewed the regulation of these processes and their involvement in human diseases like neurodegenerative disease, heart diseases, and cancers. In general, this review brings in a comprehensive review of current knowledge and provides important information for both clinical and basic investigators. In addition, the authors also added in tables to present the data in a more efficient way. However, there are several points the authors need to pay attention to improve the current version.
- The title of Table 1/3 needs to be changed, is "faults" required?
Author response: Thank you for pointing it out. We changed the title of Table 1 to ‘Examples of cancer-related UPS enzymes.’ See revised New Table 1.
- Should cite more relevant references.
Author response: As noted by the reviewer, we replaced a more relevant reference than the existing one in the revised manuscript.
- Be careful about some grammar and typo mistakes.
Author response: The revised manuscript has been carefully edited for spelling and grammatical errors.
Reviewer 3 Report
In this manuscript the authors provide an insight into the broad cellular roles of ubiquitin and ubiquitin-like proteins, touch upon some of the diseases that may be linked to the dysfunction of enzymes involved in ubiquitin metabolism, and briefly discuss clinical trials for drugs that attempt to manipulate or counter defects in ubiquitin metabolism.
The theme of the brief review is a useful and interesting one, but the current review falls short of relevant and suitably informative information. I would therefore suggest that the authors address the following points:
- A less broad title since human disease is a catchall term and cancer, neurodegenerative disease, and heart disease are only briefly covered.
- Hence, secondly, it may be more appropriate if the focus was on one of these diseases rather than trying to briefly cover all three.
- More relevant referencing is required throughout. The authors make many statements and sometimes nearly complete a paragraph without appropriate referencing of the text.
- The tables are not as informative as they could be. The authors should explain what the arrows refer to, include references to the role of the enzyme in the process (Table 1 oncogenesis, Table 3 neurodegenerative disease, Table 5, Table 7, Table 8), include references and links to the clinical trials (Table 2, Table 4, Table 6
- Section 2.1.2 – most recent clinical trials included and if so, provide a link to them. The actual drug details are missing from this section. No link to the trials or details about what stages the drugs are in, have the drugs been approved for clinical use, which are in clinical phases I-III? Lack of detail and references in this and other sections.
- Provide further detail and referencing throughout, for example, Line 187, how relevant is this study to global levels of PD? Line 191 “potential cancer progression” – how, this is not supported with relevant text or references.
Thus collectively, improve the quality of the writing by providing suitable referencing throughout and particularly concerning the tabularized data.
Author Response
Response to Reveiwers's Comments
We appreciate the reviewers’ comments and revise the manuscript accordingly. The manuscript has been improved by covering all the points of the reviewers. In this revised manuscript, changes are highlighted in red.
Reviewer number 3:
In this manuscript the authors provide an insight into the broad cellular roles of ubiquitin and ubiquitin-like proteins, touch upon some of the diseases that may be linked to the dysfunction of enzymes involved in ubiquitin metabolism, and briefly discuss clinical trials for drugs that attempt to manipulate or counter defects in ubiquitin metabolism.
The theme of the brief review is a useful and interesting one, but the current review falls short of relevant and suitably informative information. I would therefore suggest that the authors address the following points:
- A less broad title since human disease is a catchall term and cancer, neurodegenerative disease, and heart disease are only briefly covered. Hence, secondly, it may be more appropriate if the focus was on one of these diseases rather than trying to briefly cover all three.
Author response: We agree with the reviewer’s comment that cancer, neurodegenerative disease, and heart disease do not cover all human diseases. And we appreciate the suggestion that we should focus on one disease rather than all three diseases. However, cancer, neurodegenerative disease, and heart diseases are chronic diseases related to various pathological causes. One of the various pathological causes of the three chronic diseases is associated with abnormal ubiquitin and ubiquitin-like protein mechanisms. Therefore, we accepted the reviewer’s suggestion and decided to change the title of “Ub and Ubl Proteins in Human Diseases” to “Ub and Ubl Proteins in Cancer, Neurodegenerative Disorders and Heart Diseases”.
- More relevant referencing is required throughout. The authors make many statements and sometimes nearly complete a paragraph without appropriate referencing of the text. The tables are not as informative as they could be. The authors should explain what the arrows refer to, include references to the role of the enzyme in the process (Table 1 oncogenesis, Table 3 neurodegenerative disease, Table 5, Table 7, Table 8), include references and links to the clinical trials (Table 2, Table 4, Table 6)
Author response: As noted by reviewers, the revised version cited relevant literature. Each table is added more relevant examples, including links to clinical trials or literature. Also, we added an explanation of what the arrows mean in the table.
- Section 2.1.2 – most recent clinical trials included and if so, provide a link to them. The actual drug details are missing from this section. No link to the trials or details about what stages the drugs are in, have the drugs been approved for clinical use, which are in clinical phases I-III? Lack of detail and references in this and other sections.
Author response: The revised manuscript added additional information about the clinical trial and the NCT number at the reviewer's request. The new table also provided each clinical trial status along with links to websites, if available.
- Provide further detail and referencing throughout, for example, Line 187, how relevant is this study to global levels of PD? Line 191 “potential cancer progression” – how, this is not supported with relevant text or references. Thus collectively, improve the quality of the writing by providing suitable referencing throughout and particularly concerning the tabularized data.
Author response: As reviewers have pointed out several times, we have added more relevant references to the revised manuscript. We also provided additional explanations and bibliographical information for insufficient explanations pointed out by reviewers.
Round 2
Reviewer 3 Report
The authors have done an excellent job of addressing the comments for review and have put considerable effort into the writing, links and referencing for the revised manuscript. The manuscript is now suitable for publication.